# PeerJ

# Predicting changes in language skills between 2 and 3 years in the EDEN mother–child cohort

Hugo Peyre[1,2], Jonathan Y. Bernard[3,4], Anne Forhan[3,4], Marie-Aline Charles[3,4], Maria De Agostini[3,4], Barbara Heude[3,4] and Franck Ramus[1], on behalf of the EDEN Mother-Child Cohort Study Group

[1] Laboratoire de Sciences Cognitives et Psycholinguistique, Ecole Normale Supérieure, CNRS, EHESS, Paris, France
[2] Hôpital Robert Debré, Service de Psychopathologie de l'Enfant et de l'Adolescent, APHP, Paris, France
[3] Inserm, Centre for Research in Epidemiology and Population Health (CESP), Epidemiology of Diabetes and Renal Diseases Lifelong Approach Team, Villejuif, France
[4] University Paris-Sud, UMRS, Villejuif, France

## ABSTRACT

**Objective.** To examine the factors predicting changes in language skills between 2 and 3 years.

**Methods.** By using longitudinal data concerning 1002 children from the EDEN study, linear regression was used to predict 3-year language performance from 2-year language performance and the risk factors associated with language delays. Logistic regressions were performed to examine two change trajectories: children who fall below the 10th percentile of language skills between 2 and 3 years (declining trajectory), and those who rose above the 10th percentile (resilient trajectory).

**Results.** The final linear model accounted for 43% of the variance in 3-year language scores, with 2-year language scores accounting for 22%. Exposure to alcohol during pregnancy, earlier birth term, lower level of parental education and lower frequency of maternal stimulation were associated with the declining trajectory. Breastfeeding was associated with the resilient trajectory.

**Conclusions.** This study provides a better understanding of the natural history of early language delays by identifying biological and social factors that predict changes in language skills between the ages of 2 and 3 years.

Corresponding author
Hugo Peyre, peyrehugo@yahoo.fr

For some children, a very limited expressive vocabulary at 2 years is the first indication of a persistent language impairment (*Rice, Taylor & Zubrick, 2008*). Early identification of these children could lead to effective interventions to improve their social integration and academic performance (*Law, Garrett & Nye, 2003*). However, several longitudinal studies have reported that language skills in toddlerhood only poorly predict subsequent language outcome (*Bishop et al., 2003*; *Dale et al., 2003*; *Feldman et al., 2005*; *Reilly et al., 2010*; *Henrichs et al., 2011*; *Law et al., 2012*). Even when the biological and environmental

factors typically associated with language delays were added in the models, the prediction accuracy was low.

In a large Dutch sample, *Henrichs et al. (2011)* reported that the receiver operating characteristic (ROC) curve using CDI-N (Dutch version of the MacArthur-Bates Communicative Development Inventory) expressive vocabulary scores at 18 months to predict LDS (Language Development Survey; expressive vocabulary skills) delay status at 30 months had an Area Under the Curve (AUC) of 0.74, indicating only moderate predictive value. In a small study including 113 children, *Feldman et al. (2005)* reported slightly higher AUC (0.79) between CDI scores at 2 and 3 years. In the Generation R study (*Henrichs et al., 2011*), most children delayed at 18 months on the CDI-N scored in the normal range at 30 months on the LDS (positive predictive value = 29%) and most children delayed at 30 months had not scored below the 10th percentile at 18 months (sensitivity = 30%). These findings are similar to those of *Westerlund, Berglund & Eriksson (2006)*, who reported that the positive predictive value from the Swedish version of the CDI at 18 months was only 17.6% and that half the children delayed at 3 years of age had not been delayed at 18 months. *Dale et al. (2003)* and *Feldman et al. (2005)* reported higher positive predictive values (44% and 64%, respectively) and sensitivity (39% and 50%, respectively) when language delay at 2 and 3 years of age were cross-tabulated, yet more than half the children with an expressive language delay at 3 years of age in these studies had not been delayed at 2 years of age.

The picture improves only slightly when taking into account factors typically associated with language delays. In the Generation R study (*Henrichs et al., 2011*), when maternal age and education, marital status, family income, child ethnicity, parenting stress, gestational age, birth weight, child gender and age and 18-months vocabulary scores were used in a linear regression to predict LDS scores at 30 months, the model accounted for only 17.7% of the variance, with 18-months vocabulary scores accounting for 11.5%.

In the Early Language in Victoria Study (*Reilly et al., 2010*), when relying on both earlier measures of language at 2 years and the risk factors typically associated with language delays, statistical models predicted 30.4% of the variance of expressive language skills at 4 years, with late talking status at 2 years accounting for 9.5%. *Reilly et al. (2010)* also investigate the extent to which the effects of various risk factors vary across development. Biological influences on language outcomes were found to be strong at 2 years, but social disadvantage became increasingly important at age 4.

In sum, although many children show a discontinuity in the development of their language skills between 2 and 3 years, early communication was the best predictor of subsequent language functioning. Because of this discontinuity, the identification of the factors predicting trajectory changes, that is, children whose language performances vary between two time points (i.e., those who fall below the 10th percentile of language skills between two time points: the declining trajectory, and those who rose above the 10th percentile: the resilient trajectory) raise particular interest. In an article using data from children ($n = 13,016$) of the Millennium Cohort Study by *Law et al. (2012)*, children were categorized into four groups: a Typical Language (TL) group scoring within normal

limits at both times points; an Increasingly Vulnerable Language (IVL) group with a score below the norm only at the second time point; a Resilient Language (RL) group with a score below the norm only at the first time point and a Consistently Low Language (CLL) group with language delay with a score below the norm at both time points. *Law et al. (2012)* examined changes in language skills between 3 and 5 years. Among other results, that study indicated that a higher educational level of the mother was associated with the resilient trajectory (i.e., maternal education significantly distinguished CLL and RL groups). Between 18 and 30 months, the risk factors associated with the declining trajectory have been specifically studied by *Henrichs et al. (2011)*. Children in the IVL group were more likely to have mothers with younger ages and a low educational level, and to come from families with non-western parents and more parenting stress than children in the TL group. That study also reported that children in the RL group (called late bloomers) were more likely to have mothers with older ages, to come from families with non-western parents and to have lower gestational ages than children in the TL group.

In the present study, we examine the factors that predict change in language skills in a large sample of children between 2 and 3 years of age. In 2006, the US Preventive Services Task Force review examined the predictors of speech and language delays in preschool-aged children (*Nelson et al., 2006*). The most consistently reported risk factors included a family history of speech and language delay, male gender, and perinatal factors. Other risk factors reported less consistently included educational levels of the mother and father, birth order, and family size. In the present study, we also considered tobacco and alcohol consumption during pregnancy, maternal age at birth and breastfeeding because they are well established determinants of cognitive development (*Farah et al., 2008*; *O'Leary et al., 2009*; *Whitehouse et al., 2011*; *Bernard et al., 2013*).

We specifically aimed to address the following questions:

- **Question 1**: To what extent can language skills at 3 years be predicted from language skills at 2 years and from typical risk factors? As reported by previous studies, we expect language skills at 2 years to only poorly predict language skills at 3 years (sensitivity < 50%), and we expect a statistical model relying on language score at 2 years and the risk factors typically associated with language delays to predict no more than 50% of the variance of language scores at 3 years, with language scores at 2 years explaining a large amount of this variance.

- **Question 2**: What are the major risk factors associated with changes in language skills between 2 and 3 years? In particular, what are the risk factors differentiating children whose performance changes over time, i.e., showing resilient or increasingly vulnerable language, from those whose performance remains stable (consistently low and typical language respectively)?

## METHOD

### Data source

Mother–child pairs were recruited as part of the EDEN prospective mother–child cohort study (http://eden.vjf.inserm.fr). The primary aim of the EDEN cohort was to identify

prenatal and early postnatal nutritional, environmental and social determinants associated with children's health and their normal and pathological development. Pregnant women seen for a prenatal visit at the Departments of Obstetrics and Gynecology of the University Hospitals of Nancy and Poitiers before their twenty-fourth week of amenorrhea were invited to participate. Exclusion criteria were personal history of diabetes, multiple pregnancies, intention to deliver outside the university hospital or to move out of the study region within the next 3 years, and inability to speak French. The participation rate among eligible women was estimated to be 55%. Enrolment started in February 2003 in Poitiers and in September 2003 in Nancy; it lasted 27 months in each center and allowed the inclusion of 2002 women. Compared to the national perinatal survey carried out on 14,482 women who delivered in France in 2003 (*Blondel et al., 2006*), women included in the EDEN study had similar socio-demographic characteristics except that they were more educated and more often employed (details of the EDEN study protocol have been described in *Drouillet et al., 2009*).

A very broad range of data on each child's environment and development were collected from obstetrical records, questionnaire and neuropsychological tests (*Drouillet et al., 2009*).

The study was approved by the ethical research committee (Comité consultatif de protection des personnes dans la recherche biomédicale) of the Hospital of Bicêtre, and by the Data Protection Authority (Commission Nationale de l'Informatique et des Libertés). Informed written consents were obtained from the parents at enrollment for themselves and for the newborn after delivery.

Of the 2002 singleton pregnant women participating in the EDEN prospective mother–child cohort study, 1002 children were included in the analysis (Table 1 and Fig. 1).

## Variables
### Risk factors

*Child factors.* In the EDEN cohort, gender, gestational age and birth weight were collected from obstetrical records. Of the 1002 children included in the analysis, 52% were male and 48% were female. Mean (±SD) birth weight and birth term were 3.3 (±0.49) kg and 39.3 (±1.65) years respectively.

*Mother factors.* Mothers completed questionnaires on maternal age at birth and alcohol and tobacco consumption during pregnancy. In our sample, mean maternal age at birth was 29.5 (±4.7) years, 8% of the mothers reported more than 3 units of alcohol per week during pregnancy and 21% reported tobacco consumption during pregnancy.

*Family history of speech and language delay.* Mothers and fathers completed questionnaires on history of speech and language delay. 12% of the children included in the analysis had at least one parent with a history of speech and language delay.

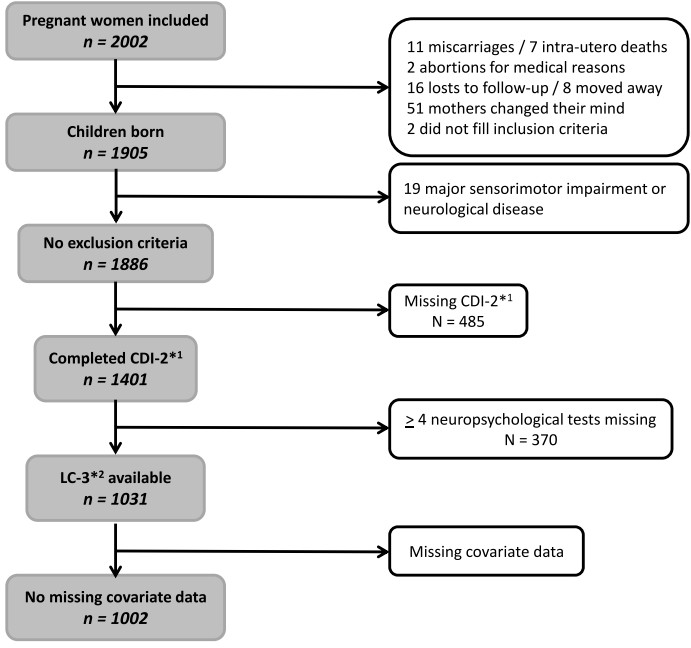

*¹ CDI-2: MacArthur Communicative Development Inventory at 2 years.
*² LC-3: Language component at 3 years.

**Figure 1 Flowchart.**

*Breastfeeding.* Of the 1002 children included in the analysis, 73% were breastfed for at least 3 days (breastfeeding initiation) and the mean duration of breastfeeding (including both partial and exclusive breastfeeding) was 4.7 (±3.7) weeks.

*Child environmental factors.* Mothers and fathers completed questionnaires on family income during pregnancy (>3000 euros/months *vs.* 2300 to 3000 euros/months *vs.* <2300 euros/months), parental education (mean of maternal and paternal school years) and the child's caretaker (mothers reported the main caretaker in the 2-year questionnaire: mother, family (father, grandparents), nursery and others (child minder, neighbor)). We further included an estimate of maternal cognitive stimulation (by averaging the weekly frequencies of storytelling, singing and playing with the child, as reported by mothers at 2 and 3 years). Birth order (number of older siblings in three classes: 0, 1 or more than 1), bilingualism (yes vs. no), the child's entry to pre-elementary school (yes vs. no; and the date of the child's entry to pre-elementary school if applicable) and the recruitment center were also considered in the analysis. Descriptive statistics of the participants are shown in Table 1.

### Language measures

*2 year-old language measure.* At 24 months of age, parents completed the short French version of the MacArthur-Bates Communicative Development Inventory [CDI-2] (*Kern, 2003*; *Kern et al., 2010*). Parents were asked to indicate from a list of 100 words if their child could say the word spontaneously (expressive vocabulary). Scores are the number

**Table 1** Summary statistics of the participating children [mean (SD) or N(%)].

| | Total sample | Typical language group | Resilient language group | Increasingly vulnerable language group | Consistently low language group |
|---|---|---|---|---|---|
| | (N = 1002) | (N = 843) | (N = 59) | (N = 59) | (N = 41) |
| **Child** | | | | | |
| Male gender, N (%) | 520 (52) | 423 (50) | 39 (66) | 33 (56) | 25 (61) |
| Birthweight, kg | 3.30 (0.49) | 3.31 (0.47) | 3.36 (0.50) | 3.17 (0.54) | 3.24 (0.58) |
| Birthterm, weeks | 39.30 (1.65) | 38.93 (2.17) | 38.69 (2.04) | 39.17 (1.99) | 39.37 (1.55) |
| **Mother** | | | | | |
| Maternal age at birth of child, years | 29.52 (4.67) | 29.49 (4.72) | 30.49 (4.41) | 28.83 (4.38) | 29.73 (4.35) |
| Alcohol during pregnancy (>3 units/week), N (%) | 79 (8) | 61 (7) | 7 (12) | 9 (15) | 2 (5) |
| Tobacco during pregnancy, N (%) | 209 (21) | 171 (20) | 13 (22) | 16 (27) | 9 (22) |
| **Family history of language delay, N (%)** | 121 (12) | 93 (11) | 8 (14) | 12 (20) | 8 (20) |
| **Breastfeeding** | | | | | |
| Initiation, N (%) | 731 (73) | 630 (75) | 44 (75) | 39 (66) | 18 (44) |
| Duration, months | 4.71 (3.74) | 4.81 (3.75) | 4.15 (3.65) | 3.87 (3.41) | 4.45 (4.20) |
| **Child's environment** | | | | | |
| Household income (euros), N (%) | | | | | |
| <2300 | 403 (40) | 318 (38) | 26 (44) | 34 (58) | 25 (61) |
| 2300–3000 | 298 (30) | 258 (31) | 14 (24) | 15 (25) | 11 (27) |
| >3000 | 301 (30) | 267 (32) | 19 (32) | 10 (17) | 5 (12) |
| Parental education, years | 13.70 (2.29) | 13.88 (2.26) | 13.42 (2.30) | 12.40 (1.92) | 14.43 (2.18) |
| Caretaker, N (%) | | | | | |
| Nursery | 226 (23) | 202 (24) | 10 (17) | 8 (14) | 6 (15) |
| Other | 468 (47) | 411 (49) | 22 (37) | 22 (37) | 13 (32) |
| Family | 94 (9) | 70 (8) | 10 (17) | 7 (12) | 7 (17) |
| Mother | 214 (21) | 160 (19) | 17 (29) | 22 (37) | 15 (37) |
| Number of older siblings, N (%) | | | | | |
| 0 | 489 (49) | 434 (51) | 21 (36) | 21 (36) | 13 (32) |
| 1 | 344 (34) | 278 (33) | 25 (42) | 22 (37) | 19 (46) |
| >1 | 169 (17) | 131 (16) | 13 (22) | 16 (27) | 9 (22) |
| Bilingualism, N (%) | 94 (9) | 82 (10) | 6 (10) | 5 (8) | 1 (2) |
| Frequency of maternal stimulation[a] | | | | | |
| between 0 and 2 years | 3.32 (0.72) | 3.35 (0.69) | 3.23 (0.80) | 3.18 (0.77) | 2.94 (0.98) |
| between 2 and 3 years | 3.19 (0.71) | 3.25 (0.67) | 2.95 (0.79) | 2.86 (0.80) | 2.83 (0.85) |
| between 0 and 3 years | 3.26 (0.61) | 3.30 (0.58) | 3.09 (0.69) | 3.02 (0.67) | 2.89 (0.81) |
| Pre-elementary schooled | | | | | |
| Yes, N (%) | 676 (67) | 584 (69) | 29 (49) | 36 (61) | 27 (66) |
| School attendance, *months* | 3.14 (3.35) | 3.18 (3.28) | 2.67 (3.71) | 2.54 (3.05) | 3.81 (4.32) |
| **Recruitement centre (Poitiers), N (%)** | 520 (52) | 434 (51) | 27 (46) | 29 (49) | 30 (73) |

*(continued on next page)*

Table 1 (*continued*)

| | Total sample | Typical language group | Resilient language group | Increasingly vulnerable language group | Consistently low language group |
|---|---|---|---|---|---|
| | (*N* = 1002) | (*N* = 843) | (*N* = 59) | (*N* = 59) | (*N* = 41) |
| **Language measures** | | | | | |
| CDI-2 | 61.03 (28.80) | 69.88 (24.21) | 13.27 (5.66) | 41.19 (17.36) | 10.80 (5.52) |
| LC-3 | 0.01 (1.01) | 0.28 (0.78) | −0.46 (0.59) | −1.84 (0.41) | −2.16 (0.54) |
| Semantic fluency | 6.83 (3.94) | 7.54 (3.67) | 4.87 (3.31) | 1.79 (2.07) | 1.44 (1.59) |
| Word and nonword repetition | 7.60 (3.24) | 8.37 (2.64) | 5.48 (3.37) | 2.89 (2.31) | 1.68 (1.98) |
| Sentence comprehension | 8.63 (2.96) | 9.17 (2.61) | 8.17 (2.46) | 4.46 (2.49) | 4.10 (2.26) |
| Sentence repetition | 7.21 (3.35) | 7.80 (3.12) | 5.75 (2.69) | 3.23 (2.08) | 2.38 (1.75) |
| Picture naming | 7.04 (1.32) | 7.40 (1.52) | 6.92 (1.29) | 4.31 (1.67) | 3.73 (1.78) |

**Notes.**

[a] On a scale of 1 (shared activities less than once per week) to 5 (shared activities nearly every day). The frequency of maternal stimulation between 0 and 3 years correspond to the average of this measure between 0 and 2 years and between 2 and 3 years.

CDI-2, MacArthur Communicative Development Inventory at 2 years; LC-3, Language component at 3 years; SD, Standard deviation.

of words produced by the child. The psychometric properties of the short French version of the MacArthur-Bates Communicative Development Inventory at 24 months have been analyzed by *Kern et al. (2010)*, showing high test–retest reliability and strong associations with the corresponding scores from the complete version.

*3 year-old language measures.* Trained psychologists individually assessed each child at 3 years by using neuropsychological tests from the ELOLA (*Evaluation du Langage Oral de L'enfant Aphasique*) (*De Agostini et al., 1998*) and NEPSY (*A Developmental NeuroPSYchological Assessment*) (*Kemp, Kirk & Korkman, 2001*; *Korkman, Kirk & Kemp, 2003*) batteries.

Five tests were used:

- **Semantic fluency** (ELOLA), which was scored as the sum of the number of animals named in one minute and of the number of objects named in one minute. As can be seen in Table 1, the mean score on the Semantic fluency test was 6.83 words (±3.94).
- **Word and nonword repetition** (ELOLA), scored as the number of words (6 items) and nonwords (6 items) repeated correctly. In our sample, the mean score on the Word and nonword repetition test was 7.60 words (±3.24).
- **Sentence comprehension** (NEPSY), a sentence comprehension task requiring pointing at one amongst 8 pictures, was scored as the number of correct answers (13 items, e.g., "montre-moi un grand lapin" ["show me a large rabbit"]). The mean score on this test was 8.63 (±2.96).
- **Sentence repetition** (NEPSY) scored as the number of sentences (17 items, e.g., "dors bien" ["sleep well"]) repeated correctly. The mean score on this test was 7.21 (±3.35).
- **Picture naming** (ELOLA), scored as the number of pictures named correctly (10 items, e.g., "cheval" ["horse"]). The mean score on this test was 7.04 (±1.32).

Since an exploratory factor analysis of the 5 variables yielded a single factor (first factor eigenvalue = 2.63; second factor eigenvalue = 0.65) explaining 53% of the total variance and having similar loadings on all variables (Semantic fluency = 0.52, Word and nonword repetition = 0.50, Sentence comprehension = 0.53, Sentence repetition = 0.57, Picture naming = 0.52), a single language component (LC-3) representing language skills at 3 years was calculated as the mean of the five scores (each score was first converted into a z-score in order for each test to have the same weight) (Table 1). The skewness (0.37) and kurtosis (0.17) of LC-3 indicate a normal distribution.

LC-3 was calculated if the number of missing scores was less than or equal to three ; this inclusive strategy is justified by the fact that for three out of five tests, the data were not missing at random, i.e., children with missing scores had significantly lower language performance (similar results were found by *Mäntynen et al., 2001*).

Because measures were not taken exactly on each child's birthday (mean age: 24.26 ± 0.81 and 38.05 ± 0.81 months for CDI-2 and LC-3 respectively), both scores were linearly corrected for the actual age of the child (children had an average increase of about 5 words per month on the 2 year-old language measure and 0.16 standard deviation per month on the 3 year-old language measure).

We defined children as being language-delayed if they were below the 10th percentile (on CDI-2 or on LC-3); this arbitrary cut-off is in line with previous research (*Dale et al., 2003*; *Henrichs et al., 2011*).

## Exclusion criteria

The population includes all children without previously known conditions associated with speech and language delay, such as hearing and neurological impairments (Fig. 1).

### *Attrition analysis of the children without exclusion criteria (see Table S1)*

In this longitudinal study, the attrition rates were 29% at 2 years and 36% at 3 years.

Similar attrition and missing data rates were reported by *Henrichs et al. (2011)*. In the Generation R Study, 29% of the children who had vocabulary scores at 2 years had missing language scores at 30 months; in our study this rate was 26%.

Compared to the 1031 children whose language scores at both ages were available, the 370 children with LC-3 only missing (due to attrition as well as other mechanisms) differed in several determinants of language skills. In particular, they were more likely to have a family history of language delay ($p < 0.001$) and they were less likely to attend school at 3 years ($p < 0.001$); Moreover, they showed significantly lower language skills at 2 years ($p = 0.004$).

Compared to the 1401 children whose language scores at 2 years were available (CDI-2), the 485 children who had missing CDI-2 also differ in several determinants of language skills. In particular, their mothers were younger ($p < 0.001$), more likely to smoke during pregnancy ($p < 0.001$), and their parents had lower family income ($p < 0.001$) and lower educational level ($p < 0.001$). Moreover, they were less likely to be the eldest child of the family and had a lower frequency of maternal stimulation during the first 2 years.

In the Generation R Study, *Henrichs et al. (2011)* also found evidence of some selective bias due to missing data (e.g., compare to children with vocabulary scores at both ages, children who had missing language scores at 30 months were more likely to have a lower birth weight and less likely to have mothers with high levels of education).

## Statistical analysis

*Question 1: Prediction of language skills at 3 years.*

First, languages measures (CDI-2 and LC-3) were analyzed as quantitative variables. An estimation of the coefficient of determination ($R^2$) was conducted in three linear regression models: models A1 and A2: with CDI-2 and LC-3 (respectively) as the dependent variable and the risk factors as independent variables; model A3: with LC-3 as the dependent variable and CDI-2 and the risk factors as independent variables.

Second, prediction of language delay at 3 years given language level at 2 years was assessed by examining sensitivity and specificity (CDI-2 as binary variable) and area under the ROC curve (CDI-2 as continuous variable).

*Question 2: Risk factors associated with changes in language skills between 2 and 3 years.*

First, the risk factors associated with changes in language skills were examined in the model A3 described above.

Second, four patterns of change were determined following *Law et al. (2012)*: a Typical Language group (TL) scoring within normal limits at both 2 and 3 years; an Increasingly Vulnerable Language (IVL) group with typical development at 2 years but language delay by 3 years; a Resilient Language (RL) group with language delay at 2 years but not anymore at 3 years; and a Consistently Low Language (CLL) group with language delay at both time points. Logistic regressions were performed to examine risk factors associated with two change profiles (RL group compared to CLL group [model B1] and IVL group compared to TL group [model B2]). Variables that showed some evidence ($p < 0.15$) of univariate association with the change trajectories were entered into the multiple logistic regression models. The significance threshold for removing variables was set at 0.15 (backward stepwise selection).

All statistical analyses were performed using SAS 9.2 software (SAS Institute, Cary, NC).

## RESULTS

*Question 1:* Table 2 shows the results of the regression analyses for the CDI-2 (model A1) and the LC-3 (model A2). Variance explained by risk factors increased slightly between 2 years (15.6%) [model A1] and 3 years (21%) [model A2], suggesting that these risk factors helped explain more variation in language skills at 3 years than at 2 years. The addition of CDI-2 to model A2 increased the variance of LC-3 explained from 21% to 43.4% [model A3]. Factors associated with both CDI-2 and LC-3 included gender, breastfeeding initiation, birth term, child's caretaker and frequency of maternal stimulation. Alcohol consumption was also significantly associated with CDI-2 only; family history of language delay; parental education and pre-elementary schooling were associated with LC-3 only.

**Table 2  Factors predicting language performance at 2, 3 and between 2 and 3 years ($N = 1002$).**

| | Model A1 | | Model A2 | | Model A3 | |
|---|---|---|---|---|---|---|
| **Dependent variable** | **CDI-2** | | **LC-3** | | **LC-3** | |
| **Independent variables** | **Risk factors** | | **Risk factors** | | **CDI-2 and risk factors** | |
| | $R^2 = 15.6\%$ | | $R^2 = 21.0\%$ | | $R^2 = 43.4\%$ | |
| | $\beta$ | $p$ | $\beta$ | $p$ | $\beta$ | $p$ |
| **Child** | | | | | | |
| Male gender | −7.55 | <0.001 | −0.30 | <0.001 | −0.16 | 0.001 |
| Birthweight, kg | 2.53 | 0.3 | 0.12 | 0.1 | 0.09 | 0.2 |
| Birthterm, weeks | 1.81 | 0.005 | 0.05 | 0.03 | 0.01 | 0.47 |
| **Mother** | | | | | | |
| Maternal age at birth of child, years | −0.12 | 0.6 | 0.01 | 0.3 | 0.01 | 0.08 |
| Alcohol during pregnancy (>3 units/week) | −6.07 | <0.001 | −0.13 | 0.2 | −0.03 | 0.8 |
| Tobacco during pregnancy | 1.11 | 0.6 | −0.01 | 0.9 | −0.01 | 0.6 |
| **Family history of language delay** | −4.98 | 0.06 | −0.19 | 0.03 | −0.10 | 0.2 |
| **Breastfeeding initiation** | 4.86 | 0.02 | 0.15 | 0.03 | 0.06 | 0.3 |
| **Child's environment** | | | | | | |
| Household income (euros) | | 0.3 | | 0.1 | | 0.04 |
|   <2300 | −2.22 | | −0.16 | | −0.11 | |
|   2300–3000 | 1.40 | | −0.16 | | −0.17 | |
|   >3000 | ref. | | ref. | | ref. | |
| Parental education, years | 0.75 | 0.1 | 0.09 | <0.001 | 0.08 | <0.001 |
| Caretaker | | <0.001 | | 0.006 | | 0.4 |
|   Nursery | 6.86 | | 0.12 | | 0.00 | |
|   Other | 8.41 | | 0.20 | | 0.06 | |
|   Family | −1.02 | | −0.12 | | −0.09 | |
|   Mother | ref. | | ref. | | ref. | |
| Number of older siblings | | 0.2 | | 0.1 | | 0.2 |
|   0 | ref. | | ref. | | ref. | |
|   1 | −3.89 | | −0.11 | | −0.04 | |
|   >1 | −2.68 | | −0.18 | | −0.15 | |
| Bilingualism | 2.89 | 0.3 | 0.05 | 0.6 | 0.00 | 1 |
| Frequency of maternal stimulation[a] | | | | | | |
|   between 0 and 2 years | 8.79 | <0.001 | – | – | – | – |
|   between 2 and 3 years | – | – | – | – | 0.09 | 0.02 |
|   between 0 and 3 years | – | – | 0.27 | <0.001 | – | – |
| Pre-elementary schooled | – | – | 0.14 | 0.03 | 0.09 | 0.09 |
| **Recruitement centre** (Poitiers) | −0.35 | 0.8 | −0.03 | 0.6 | −0.02 | 0.7 |
| **Language measures** | | | | | | |
| CDI-2 | – | – | – | – | 0.02 | <0.001 |

**Notes.**

[a] On a scale of 1 (shared activities less than once per week) to 5 (shared activities nearly every day). The frequency of maternal stimulation between 0 and 2 years was used in model A1, between 0 and 3 years in model A2 and between 2 and 3 years in model A3.

CDI-2, MacArthur Communicative Development Inventory at 2 years; LC-3, Language component at 3 years; $\beta$, Regression coefficient with 95% confidence interval (95% CI); $p$, $p$-value; $R^2$, Coefficient of determination.

**Table 3** Factors associated with the resilient trajectory (model B1: Resilient Language group vs. Consistently Low Language group) and the declining trajectory (model B2: Increasingly Vulnerable Language group vs. Typical Language group).

| | Model B1 | | | Model B2 | | |
| | Resilient language vs Consistently low language | | | Increasingly vulnerable language vs Typical language | | |
| --- | --- | --- | --- | --- | --- | --- |
| | OR | 95% CI | $p$ | OR | 95% CI | $p$ |
| Alcohol during pregnancy (>3 units/week) | – | – | – | 2.29 | 1.04–5.02 | **0.04** |
| Breastfeeding initiation | 3.75 | 1.60–8.78 | **0.002** | – | – | – |
| Birth term, *weeks* | – | – | – | 0.81 | 0.71–0.93 | **<0.001** |
| Parental education, *years* | – | – | – | 0.76 | 0.67–0.87 | **<0.001** |
| Frequency of maternal stimulation[a] | – | – | – | 0.52 | 0.37–0.75 | **0.001** |

Notes.

[a] On a scale of 1 (shared activities less than once per week) to 5 (shared activities nearly every day). The variable frequency of maternal stimulation between 2 and 3 years was used in models B1 and B2.

OR, Odds ratios with 95% confidence interval (95% CI); $p$, $p$-value (in bold if $p < 0.05$).

Sensitivity of the CDI-2 to predict delayed children at 3 years was 41.0%, the positive predictive value of the CDI-2 to predict delayed children at 3 years was also 41.0% and the AUC = 0.85 (95% CI: 0.81–0.88).

*Question 2:* While most risk factors had a significant influence on language skills at 2 and at 3 years [models A1 and A2], only some risk factors had an influence between 2 and 3 years: once language level at 2 years was known, only male gender ($p = 0.001$), income ($p = 0.04$), level of parental education ($p < 0.001$) and frequency of maternal stimulation ($p = 0.02$) explained additional variance at 3 years [model A3].

There were 41 children in the Consistently Low Language (CLL) group, 59 in the Resilient Language (RL) group, 59 in the Increasingly Vulnerable Language (IVL) group and 843 in the Typical Language (TL) group (Table 1). Model B1 indicated that only breastfeeding distinguished between RL and CLL groups (OR = 3.75; IC-95% [1.60–8.78]; $p$-value = 0.002) (Table 3). In the RL group 75% were breastfed ($n = 44$; among these children, the mean breastfeeding duration = 4.15 ($\pm3.35$)) whereas in the CLL group, only 44% were breastfed ($n = 18$; mean breastfeeding duration = 4.69 ($\pm4.19$). In model B2, alcohol consumption during pregnancy (OR = 2.29; IC-95% [1.04–5.02] ; $p = 0.04$), birth term (OR = 0.81; IC-95% [0.71–0.93]; $p < 0.001$), parental education (OR = 0.76; IC-95% [0.67–0.87]; $p < 0.001$) and frequency of maternal stimulation (OR = 0.52; IC-95% [0.37–0.75]; $p = 0.001$) significantly differentiated between IVL and TL groups.

## DISCUSSION

With respect to our first question, our ability to predict language skills at 3 years remains limited. Indeed, linear models including a measure of language at 2 years and the main risk factors predict only 43% of the variance of language at 3 years, with CDI-2 scores at 2 years accounting for 22%. Our estimates of the variance explained are higher than those found in the Generation R study (the model accounted for only 18% of the variance of LDS scores

at 30 months, with 18-months vocabulary scores accounting for 12%) and in the Early Language in Victoria Study (30% of the variance of expressive language skills at 4 years was explained by the model, with late talking status at 2 years accounting for 10%), but this may be at least partly explained by the shorter lag between the two time points in our study. Although language skills at 2 years have a fair predictive power of language skills at 3 years, more than half the children with an expressive language delay at 3 years of age had not been delayed at 2 years of age (sensitivity = 41%). Regarding our ability to predict language skills at 3 years from language skills at 2 years, the sensitivity, positive predictive value (41%) and AUC (0.85) were similar to those of *Feldman et al. (2005)*.

In line with those studies, the best predictor of language functioning was found to be early vocabulary (Question 2). Changes in language skills between the ages of 2 and 3 years was influenced by gender (−0.16 SD in male) and several factors related to the child's environment: the level of parental education (+0.08 SD *per year*), income (−0.11 SD and −0.17 SD for incomes <2300 and between 2300 and 3000 euros/months compared to income >3000 euros/months) and the frequency of maternal stimulations (+0.09 SD *per unit*).

We also identified risk (and protective) factors differentiating children whose performance changed over time, i.e., showing resilient or increasingly vulnerable language, compared to those whose performance remains stable (consistently low and typical language respectively). Children who showed a declining trajectory between 2 and 3 years had increased exposure to alcohol during pregnancy, lower level of parental education, earlier birth term and lower frequency of maternal stimulation. On the other hand, breastfeeding increased the likelihood of having a resilient trajectory. These are all well-known determinants of cognitive development among young children. Less linguistically rich environments have been consistently associated to poorer child language outcomes (*Farah et al., 2008*). In the Generation R study, *Henrichs et al. (2011)* also reported that children who showed a declining trajectory were more likely to have mothers with a low educational level. The effect of alcohol consumption during pregnancy (depending on dose, duration, and pattern of drinking) on the cognitive development of the child is well supported by the scientific literature (*Larroque et al., 1995*; *O'Leary et al., 2009*). Even in children born after 37 weeks (only 2.2% of the children were born preterm in our sample), associations between birth term and cognitive development have been reported (*Yang, Platt & Kramer, 2010*). Many studies have shown that breastfeeding was associated with better language skills in children (*Whitehouse et al., 2011*; *Bernard et al., 2013*). In our study, the effects of alcohol consumption during pregnancy, birth term and breastfeeding seem to be partly delayed. These results contrast with the intuitive idea that such biological factors show mostly early influences (i.e., up to 2 years), and that social factors rather have later influences. In fact it is perfectly possible that some biological factors may show increasing effects when language abilities become more elaborated (i.e., between 2 and 3 years).

The strengths of the EDEN study include its longitudinal design with repeated measurements of language development. Although the language measure at 2 years was

based solely on parental report of expressive vocabulary, the measures at 3 years were made by trained clinicians using several tests tapping multiple relevant dimensions of language (vocabulary, phonology, syntax). Whereas the richness of the 3-year-old measures is a strength, the qualitative differences between language measures at the two time points is a limitation, thus estimates of the increase of variance explained by risk factors between 2 and 3 years needs to be interpreted cautiously. There was also evidence for some selective bias due to missing data; indeed, children whose data were available at both ages had better language skills at 2 years and fewer risk factors for language delay than children with missing LC-3 or CDI-2 (see Table S1). Such a bias reduces the variance of our sample and therefore the statistical power of our analysis.

Our ability to predict which toddlers have language delay at 3 years remains modest at best. As language skills are still unstable after 3 years (as shown by the study of *Law et al., 2012*), analyses of language measures acquired in the EDEN study at 5 years and later ages will be important to further refine our understanding of the trajectories of language development. Future research is also needed to identify the age period in which population-wide screenings for language problems are the most useful.

### Abbreviations

| | |
|---|---|
| **CDI-2** | MacArthur-Bates Communicative Development Inventory at 2 years |
| **EDEN** | Etude des Déterminants pré et postnatals précoces du développement et de la santé de l'ENfant |
| **LC-3** | Language skills at 3 years |

## ACKNOWLEDGEMENTS

We are indebted to the participating families, the midwife research assistants (L Douhaud, S Bedel, B Lortholary, S Gabriel, M Rogeon, and M Malinbaum) for data collection, the psychologists (Marie-Claire Cona and Marielle Paquinet) and P Lavoine, J Sahuquillo and G Debotte for checking, coding, and data entry. We also thank S Kern for providing the French version of the MacArthur Communicative Development Inventory.

Members of the EDEN mother–child cohort study group are as follows: MA Charles, M De Agostini, A Forhan, B Heude and P Ducimetière (INSERM, UMR-S 1018, Team 10); M Kaminski, MJ Saurel-Cubizolles, P Dargent, X Fritel, B Larroque, N Lelong, L Marchand, and C Nabet (INSERM, U953); I Annesi-Maesano (INSERM U707); R Slama (INSERM, U953); V Goua, G Magnin and R Hankard (Poitiers University Hospital); O Thiebaugeorges, M Schweitzer, B Foliguet (Nancy University Hospital); and N Job-Spira (ANRS).

### Funding

This EDEN study was supported by: Fondation pour la Recherche Médicale (FRM), French Ministry of Research IFR and Cohort Program, INSERM Nutrition Research Program,

French Ministry of Health Perinatality Program, French Agency for Environment Security (AFFSET), French National Institute for Population Health Surveillance (INVS), Paris-Sud University, French National Institute for Health Education (INPES), Nestlé, Mutuelle Générale de l'Education Nationale (MGEN), French Speaking Association for the Study of Diabetes and Metabolism (Alfediam), National Agency for Research (ANR non thematic program), and National Institute for Research in Public Health (IRESP TGIR Cohorte Santé 2008 Program). Additional funding came from ANR contracts ANR-10-LABX-0087, ANR-11-0001-02 PSL*, and ANR-12-DSSA-0005-01. The funders had no role in study design, data collection and analysis, decision to publish, or preparation of the manuscript.

**Grant Disclosures**

The following grant information was disclosed by the authors:

Fondation pour la Recherche Médicale (FRM).

French Ministry of Research IFR and Cohort Program.

INSERM Nutrition Research Program.

French Ministry of Health Perinatality Program.

French Agency for Environment Security (AFFSET).

French National Institute for Population Health Surveillance (INVS).

Paris-Sud University, French National Institute for Health Education (INPES).

Nestlé, Mutuelle Générale de l'Education Nationale (MGEN).

French Speaking Association for the Study of Diabetes and Metabolism (Alfediam).

National Agency for Research (ANR non thematic program).

National Institute for Research in Public Health (IRESP TGIR Cohorte Santé 2008 Program).

ANR contracts: ANR-10-LABX-0087, ANR-11-0001-02 PSL*, ANR-12-DSSA-0005-01.

**Competing Interests**

The authors declare they have no competing interests.

**Author Contributions**

- Hugo Peyre analyzed the data, contributed reagents/materials/analysis tools, wrote the paper, prepared figures and/or tables, reviewed drafts of the paper.
- Jonathan Y. Bernard analyzed the data, contributed reagents/materials/analysis tools, reviewed drafts of the paper.
- Anne Forhan, Marie-Aline Charles, Maria De Agostini and Barbara Heude conceived and designed the experiments, performed the experiments, reviewed drafts of the paper.
- Franck Ramus analyzed the data, wrote the paper, reviewed drafts of the paper.

**Human Ethics**

The following information was supplied relating to ethical approvals (i.e., approving body and any reference numbers):

The study was approved by the ethical research committee (Comité consultatif de protection des personnes dans la recherche biomédicale) of the Hospital of Bicêtre, and by

the Data Protection Authority (Commission Nationale de l'Informatique et des Libertés). Informed written consents were obtained from the parents at enrollment for themselves and for the newborn after delivery.

## Supplemental Information

Supplemental information for this article can be found online at http://dx.doi.org/10.7717/peerj.335.

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
