# Peer review of "Predicting changes in language skills between 2 and 3 years in the EDEN mother–child cohort"

_PeerJ, doi:10.7717/peerj.335_

## Round 0.1 · original submission · Minor Revisions

Both reviewers were generally positive about the study design and findings, but indicated that certain information was missing or confusing in the submitted version. Both have also made specific suggestions for improvements and clarifications in the text. Please carefully consider the comments of each reviewer in your revised manuscript. It would be extremely helpful if your revision highlights the changes that you make in response to the reviewers' comments (e.g., bold, italics etc.).

Please also note that the revised manuscript may be re-reviewed by one or both reviewers. Thank you again for your submission .

·

Basic reporting

The manuscript is generally well written. The background to the study is well presented. While importance is not one of the editorial criteria, the study does make a very useful contribution to what is known about patterns and predictors of change in typical and atypical language abilities from 2-3 years. The readability of the manuscript could be improved by rearranging some of the text. In the Introduction, Law et al.’s (2012) study is first discussed on line 99 (the age of the children needs to be included here) and then again on line 110. I suggest bringing this information together in the same paragraph. The variables associated with typical and atypical language development in the first three years of life (i.e., the rationale for inclusion of these variables in the current study) need to be discussed in the ‘Introduction’. Currently, the rationale for the independent variables is included in the ‘Methods’. A related issue is that information that belongs in the ‘Introduction’ is included on lines 119-124 following the first research question. The background to the research questions should lead up to the research questions and follow the research questions. Not all the literature cited is appropriately referenced and all the citations need to be checked for accuracy (e.g., Zubrick et al.’s. (2007) study is incorrectly cited as a study of language outcomes of late talkers). I have a query about the citations for the European French version of the CDI-2 (100 words) at 24 months. The children in Kern’s (2007) study were much younger than children in this study and the children in Reilly et al.’s (2007) study completed an Australian adaptation of the English version of the CDI: Words and Sentences (680 words). Citations for the European French version of the CDI-2 need to be included here. In the ‘Results’ it is puzzling why only 3 characteristics of the participants were discussed in the text (i.e., CDI-2 score, family history, bilingualism). It would be useful if summary statistics for the 4 groups of children: ‘Consistently TL’, ‘IVL’, ‘RL’ and ‘CLL’ were included in Table 1. The names of these 4 groups are not sufficiently intuitive or well known to justify the use of the abbreviations in the text and table titles. The names of the 4 change patterns rather than abbreviations should be used where practical to do so.

Experimental design

The data source is the EDEN prospective mother-child Cohort Study. Data on the participation rate of women on entry to this cohort should be provided. In addition, the characteristics of the cohort need to be compared with the characteristics of the reference population. Sample attrition is well described. Information about access and use of data from the EDEN Cohort needs to be included under the heading ‘Data source’. The research questions could be simplified and numbered for ease of reference and reiterated in the ‘Statistical analysis’ and ‘Discussion’ sections. The risk factors under investigation in this study were referenced to a 2006 systematic evidence review (Nelson et al., 2006). Epidemiological evidence about risk factors for early language delay (as discussed in the Introduction) is the appropriate source of data to justify the choice of risk variables for this study. As stated earlier, the rationale for the risk variables belongs in the Introduction. In the text, the risk variables need to be grouped under the same headings used in the tables and discussed in a logical order (e.g., child, maternal, family factors). The levels of each of the independent variables and the reference category for each variable should be documented in the Method and this should be consistent with the reporting of the variables in the tables. Child age is an important variable and warrants more detail about the correction for age.

Validity of the findings

The parameterization of the independent variables in the analyses (Model A1, A2, A3, B1 and B2) needs to be checked in the text (and the tables) and explained under the heading ‘Statistical analysis’. For instance, why is the reference category for ‘Number of older siblings’ is ‘>1’ instead of ‘no siblings’ (i.e., ‘0’), which represents the lowest level of risk. The ‘Breastfeeding initiation’ variable needs further explanation (e.g., what was the time period for initiation of breastfeeding?).There is insufficient information about the two dependent variables: CDI-2 and the ELOLA. There is no information about the psychometric properties of the CDI-2 and the ELOLA and this needs to be detailed. The rationale for choosing a score below the 10th percentile on the CDI-2 as the criterion for identifying children with language delay at 2 years needs to be included under the heading ‘2-year-old measure’. More information about the ELOLA is needed. For example, is this a standardised omnibus test that yields standard scores for each subtest and a standard score for overall performance on the test? The rationale for calculating z scores for the ELOLA needs to be provided. In the ‘Abstract’ and the ‘Discussion’ the direction of the associations between independent and dependent variables needs to be specified (e.g., ‘lower frequency of maternal stimulation’, ‘higher level of parental education’ etc.).

·

Basic reporting

Thank you for the opportunity to review this manuscript. The manuscript presents on the interesting topic of predicting language skills in young children and predicting change in language skills between 2 – 3 years old. Factors that may influence early language skills and the trajectory of language impairment are current topics of interest amongst several groups of researchers. The results of this study add to this body of literature and are in line with previous findings such as those of Henrichs et al. (2011) and Reilly et al. (2010).

Experimental design

The research questions were clear and similar to questions posed by previous research, but nonetheless meaningful and with a different longitudinal dataset. It was helpful that the authors used the same categorization groups as Law et al. (2012) to examine trajectories of change as was presenting the attrition rate analysis. It might be useful to comment on the attrition rate in the current study relative to previous published research.

The language measure at 2 years is markedly different to the language measures used at 3 years. Perhaps this is one possible reason why language skill at 2 years is limited in predicting language skill at 3 years. Having said that though, the authors do identify this issue in the discussion.

The analysis is appropriate to the research questions posed.

Validity of the findings

The discussion concerning the ‘delayed’ effects of biological factors (alcohol during pregnancy, birth term and breastfeeding) was interesting but I was left wanting a further explanation from the authors as to why biological factors show "increasingly evident effects as cognitive abilities become increasingly sophisticated".

Additional comments

The paper is well written and the study informative.
I do wonder whether the abstract's conclusion needs to be further explained re: "this study provides a better understanding of the natural history of early language delays". A better understanding relative to what? How has this study provided a better understanding?

---

## Round 0.2 · Minor Revisions

I appreciate the effort you have made to address the comments of both reviewers. While I believe the manuscript is publishable in its current form, I strongly urge you to incorporate the suggestions provided by Reviewer 1 regarding this version of the manuscript. Provided these suggested amendments are addressed, I will issue a final decision without the need for further review, and I look forward to your re-submission.

·

Basic reporting

The revised manuscript meets the reporting requirements.

Experimental design

The study sample comprised 1002/2002 children in the EDEN prospective mother-child cohort. This information needs to be included under the heading ‘Data source’ on p. 6 and not under the subheading ‘Breastfeeding’ on page 8. The authors state that mothers in the EDEN prospective mother-child cohort differed from the reference population on maternal education and maternal employment. This is not unusual in cohort studies. I suggest that the authors delete the statement ‘Our study population is not representative of the general population’ on p. 7. Are there other comparisons that allow the authors to state that the EDEN cohort is broadly representative of the reference population, except for higher levels of maternal education and employment? It would be sufficient to make a statement about this and cite references that describe the EDEN cohort in detail. Drouillet et al. (2009) state that the participation rate is ‘55%’ not ‘about 55%’.

Validity of the findings

The revised manuscript meets the appropriate standards.

---

## Round 0.3 · accepted · Accept

Thank you for the careful attention to the suggestions made by both reviewers.